Using vessels of opportunity for determining important habitats of bottlenose dolphins in Port Phillip Bay, south-eastern Australia

Ledwidge Maddison J. 1 m.ledwidge@deakin.edu.au
http://orcid.org/0000-0002-1874-0619 Monk Jacquomo 2
http://orcid.org/0000-0002-6053-9062 Mason Suzanne J. 1
http://orcid.org/0000-0003-1124-9330 Arnould John P. Y. 1
1 School of Life and Environmental Sciences, Deakin University , Burwood, Victoria , Australia
2 Institute for Marine and Antarctic Studies, University of Tasmania , Hobart, Tasmania , Australia
Pochon Xavier
Electronic publication date: 2024 Oct 30
Publication date: 2024
Volume: 12
Electronic Location ID: e18400
Received 2023 Nov 8; Accepted 2024 Oct 4
Copyright: © 2024 Ledwidge et al.
Copyright year: 2024
Copyright holder: Ledwidge et al.
License: This is an open access article distributed under the terms of the Creative Commons Attribution License, which permits unrestricted use, distribution, reproduction and adaptation in any medium and for any purpose provided that it is properly attributed. For attribution, the original author(s), title, publication source (PeerJ) and either DOI or URL of the article must be cited.
License URL: https://creativecommons.org/licenses/by/4.0/

Keywords: Tursiops, Habitat suitability, Conservation management, Species distribution, Ecotourism, Ensemble model

Funding: The authors received no funding for this work.

==============================
Understanding species’ critical habitat requirements is crucial for effective conservation and management. However, such information can be challenging to obtain, particularly for highly mobile, wide-ranging species such as cetaceans. In the absence of systematic surveys, alternative economically viable methods are needed, such as the use of data collected from platforms of opportunity, and modelling techniques to predict species distribution in un-surveyed areas. The present study used data collected by ecotourism and other vessels of opportunity to investigate important habitats of a small, poorly studied population of bottlenose dolphins in Port Phillip Bay, south-eastern Australia. Using 16 years of dolphin sighting location data, an ensemble habitat suitability model was built from which physical factors influencing dolphin distribution were identified. Results indicated that important habitats were those areas close to shipping channels and coastlines with these factors primarily influencing the variation in the likelihood of dolphin presence. The relatively good performance of the ensemble model suggests that simple presence-background data may be sufficient for predicting the species distribution where sighting data are limited. However, additional data from the center of Port Phillip Bay is required to further support this contention. Important habitat features identified in the study are likely to relate to favorable foraging conditions for dolphins as they are known to provide feeding, breeding, and spawning habitat for a diverse range of fish and cephalopod prey species. The results of the present study highlight the importance of affordable community-based data collection, such as ecotourism vessels, for obtaining information critical for effective management.

Introduction

Individual species require varying habitat resources and conditions, such as food, shelter and breeding space, to support survival and reproduction (Gaillard et al., 2010). Environmental and anthropogenic factors such as prey distribution and habitat modification can influence the availability and quality of habitat requirements and, therefore, species distribution, ecology and fitness (Mair et al., 2014). For example, the snow leopard (Panthera uncia) only occurs in mountain ranges of central and South Asia due to the region’s high abundance of its main prey item, blue sheep (Pseudois nayaur) (Aryal et al., 2016). Similarly, in response to climate change impacts on the oceanography of the Antarctic Peninsula, changes in the abundance of Antarctic krill (Euphausia superba) in surrounding waters have caused shifts in the foraging areas of Adélie penguins (Psygoscelis adeliae) (Hinke et al., 2007).

Correspondingly, understanding the critical habitat requirements that drive species distributions and fitness is fundamental for effective conservation, planning and management (Charbonnel et al., 2022). This is particularly important for top-order predators that maintain ecosystem structure and function through regulating prey populations (Eierman & Connor, 2014). However, acquiring habitat information can present logistical challenges, especially for rare and elusive species (Zotos, Stamatiou & Vogiatzakis, 2022). Consequently, the accuracy and availability of such information can vary across taxonomic groups and geographical regions (Kaschner et al., 2012).

Traditional methods such as observational surveys, telemetric tracking (e.g., VHF, GPS, satellite) and home-range mapping, are reliable and efficient ways to assess species distribution (Hofman et al., 2019; Moxley et al., 2017). However, there are spatial and temporal limits to such survey methods, and technical challenges relating to design, construction and operation of telemetric devices can prevent consistent tracking (Gaillard et al., 2010; Hofman et al., 2019). Obtaining spatial and temporal data via these methods proves particularly challenging for elusive species, which can differ in their activity patterns and inhabit vast, remote environments (Zotos, Stamatiou & Vogiatzakis, 2022).

An alternative approach for assessing species’ distribution is habitat suitability modelling (Charbonnel et al., 2022; Lasram et al., 2020). Such models predict the suitability of a location for a species based on their relationship with environmental predictors reflecting climate, topography, oceanographic processes and prey distribution (Charbonnel et al., 2022; Elith et al., 2011). Predictions from most suitable (1) to least suitable (0) habitat provide an understanding of species niche requirements and movement patterns, which allows enhanced conservation management for a target species, community or ecosystem (Lasram et al., 2020). Model use has been especially important in evaluating ecological impacts of pollution and climate change, assessing risk of biological invasions, and endangered species management (Chai et al., 2016; Charbonnel et al., 2022). Ideally, data used in models should originate from well-designed surveys that ensure adequate coverage of target populations (Fiedler et al., 2018). However, well-designed surveys may not be possible due to cost, especially for mobile, wide-ranging species that inhabit remote areas such as offshore marine environments (Redfern et al., 2006). Consequently, in such situations, data collection relies on opportunistic sightings of population areas (Kaschner et al., 2012). As logistical and financial challenges are associated with obtaining accurate spatial data for marine mammals, which spend extensive periods of time submerged and range over vast distances, habitat suitability modelling has been widely used in assessing critical habitat requirements of cetaceans (Correia et al., 2021; Redfern et al., 2006).

The bottlenose dolphin (Tursiops spp.) is a highly mobile cetacean, globally distributed throughout temperate and tropical waters and occupying a variety of coastal and offshore habitats (Bearzi, 2023; Paschoalini & Santos, 2020). There are currently three widely accepted species in the Tursiops genus, the common bottlenose (T. truncatus), the Indo-Pacific bottlenose (T. aduncus) and Tamanend’s bottlenose (T. erebennus), which feed on a variety of prey including bony and cartilaginous fishes, cephalopods and crustaceans (Paschoalini & Santos, 2020). The habitat needs and distribution of these species are influenced by a variety of biotic and abiotic environmental parameters as well as anthropogenic factors (Chilvers, Corkeron & Puotinen, 2003; Hanf et al., 2022). Abiotic influences on bottlenose dolphin distribution include geomorphology of seafloor, and climate and physiochemical parameters which can be short-term (e.g., daily, seasonal) or long-term (e.g., yearly). Biotic influences include prey availability, interspecific competition, and predation (Hanf et al., 2022; Haughey et al., 2021). Anthropogenic influences on dolphin distribution include commercial and recreational activities, such as fishing trawlers driving bottlenose dolphin foraging to coastal waters of a small embayment, negatively impacting natural dolphin foraging behavior and social structure (Chilvers, Corkeron & Puotinen, 2003). Environmental and anthropogenic variables such as these can influence the different movement patterns and home-range sizes displayed by various sub-species and populations of bottlenose dolphins (Paschoalini & Santos, 2020). Therefore, a change in factors such as prey distribution and availability can negatively impact resident populations and those with small home-ranges (Passadore et al., 2017).

A small, poorly studied resident population of bottlenose dolphins inhabits Port Phillip Bay on the northern extent of Bass Strait, south-eastern Australia (Charlton-Robb, Taylor & McKechnie, 2015) (Fig. 1A). This distinct, spatiotemporally segregated population, along with another in the Gippsland Lakes (also in south-eastern Australia) has recently been proposed to comprise a unique newly described species, the Burrunan dolphin (T. australis) (Charlton-Robb et al., 2011). In Victoria, this nominal species is listed by the state government as ‘Critically Endangered’ (Victorian Department of Energy Environment and Climate Action, 2024) due to raised concerns about its nominally small population size, limited gene flow and increasing environmental and anthropogenic pressures (Filby et al., 2017; Filby, Stockin & Scarpaci, 2014). However, the nomination of these populations as a species has not been recognized by the Society for Marine Mammalogy (Committee on Taxonomy, 2022) and, consequently, they will hereafter be referred to as bottlenose dolphin (Tursiops sp.).

Figure 1 Sampling area and distribution of bottlenose dolphin sightings from 2006–2023 in Port Phillip Bay, south-eastern Australia.

(A) Ecotourism and other vessel of opportunity operational areas (red and green polygons) in Port Phillip Bay (map source: Esri, GEBCO, NOAA, National Geographic, Garmin, Geonames.org, and other contributors), and (B) distribution of bottlenose dolphin presence and background points for habitat suitability modelling (map source: Victorian Department of Energy Environment and Climate Action (2024)).

South-eastern Australia is one of the fastest warming marine regions in the world (Spillman et al., 2021) and the anticipated oceanographic changes are likely to impact the diversity, abundance, and distribution of prey populations important to bottlenose dolphins (Holland et al., 2020). In addition, continued coastal urban growth and industrialization in the region could have detrimental effects on large marine predators such as bottlenose dolphins (Marley et al., 2017; Zanardo et al., 2017) in Port Phillip Bay. Correspondingly, informed conservation management protocols are required to maintain the viability of this bottlenose dolphin population under such environmental pressures (Filby, Stockin & Scarpaci, 2014). However, the large area of Port Phillip Bay (~1,900 km2) makes research on the distribution of this dolphin population logistically and financially challenging. Consequently, little is known of its critical habitat requirements (Beddoe et al., 2024; Sampson, Easton & Singh, 2014).

Utilizing ecotourism vessels as platforms for the collection of cetacean occurrence data is an underused cost-effective method that has been proven to produce reliable results in the investigation of habitat affiliations with cetacean distribution (Currie et al., 2018; Embling, Walters & Dolman, 2015). In southern Port Phillip Bay, there are a variety of “swim-with-dolphin” and ecotourism boat charters that have operated daily throughout the summer months for numerous decades and opportunistically recorded dolphin sightings for over a decade (Filby, Stockin & Scarpaci, 2014; Scarpaci, Corkeron & Nugegoda, 2003). This economically viable method of data collection presents a unique opportunity to assess relationships between dolphin occurrence and environmental factors, as high data volumes are produced at a low cost (Charbonnel et al., 2022; Embling, Walters & Dolman, 2015).

Therefore, using data collected by ecotourism vessels and other vessels of opportunity in Port Phillip Bay, the aims of the present study were to: (1) develop an ensemble habitat suitability model to investigate the most important environmental factors influencing bottlenose dolphin distribution; and (2) use the model to predict their potential distribution throughout all of Port Phillip Bay.

Materials and Methods

Study area, data collection and processing

The study was conducted mostly in the southern extent of Port Phillip Bay, a large semi-enclosed marine embayment that is located on the northern coast of Bass Strait, south-eastern Australia (Sampson, Easton & Singh, 2014) (Fig. 1A). The bay is relatively shallow (mean depth 13.6 m), with a maximum depth of only 24 m in the center. The south-eastern region of the bay is characterized by steep bathymetric gradients associated with a channel that opens into Bass Strait through Port Phillip Heads (Holdgate et al., 2001). Port Phillip Bay is a busy waterway accommodating various vessel types. Each year ~3,500 cargo and container ships visit the port of Melbourne, making it Australia’s largest container and general cargo port (Sampson, Easton & Singh, 2014). Numerous recreational fishing boats, cruise ships and ecotourism vessels operate within Port Phillip Bay, alongside two passenger ferries and eight commercially licensed longline fishing vessels (Sampson, Easton & Singh, 2014; Victorian Fisheries Authority, 2023). Sightings of dolphins by staff and patrons onboard licensed ecotourism vessels within the south-eastern region of the bay were recorded opportunistically by the operators throughout the austral spring-autumn period (September to April), with these collated records forming the basis for the present study. In addition, data were obtained from the publicly available Victorian Biodiversity Atlas (VBA) database (Victorian Department of Energy Environment and Climate Action, 2024) which includes sightings from vessels of opportunity and researchers. Where multiple sightings were recorded on the same day, only those sightings with a minimum time interval of >15 min or distance >500 m were considered unique presence records. Thresholds were selected based on methods of Lacetera et al. (2023), and DEECA vessel regulations regarding marine mammal approach limits in Port Phillip Bay (Hale, 2002). To identify potential bias associated with repetitive sampling of individuals, distance between unique presence records was further examined. On the same day <20% of all records were within a 500 m distance and were therefore included in further analyses.

Since 2006, information on sightings (Global Position System (GPS) derived latitude and longitude; date and time) have been obtained from vessel logbooks, the VBA and the metadata from digital images of dolphin dorsal fins taken by patrons on ecotourism vessels. Fine-scale data collection methods (e.g., route, speed and duration of vessel tours; sea-state conditions etc.) could not be ascertained accurately for these data. However, all records used in the present study were obtained from vessels that adhered to DEECA regulations regarding marine mammal approach limits for tour operators and research vessels. These included a speed <10 knots (18.5 km·h−1) being maintained when <100 m from a dolphin or pod, and remaining a maximum of 20 min when within a 100 m perimeter (Hale, 2002). In addition, tours are limited by the weather for safety, patron comfort and dolphin visibility, operating only in Beaufort Sea State Conditions <3. Nonetheless, due to the uncertainty of search methods and effort, these data were used in presence-background analyses of habitat affiliations with dolphin distribution.

Environmental variables

Processing of environmental data was conducted using ArcGIS (Version 10.8; ESRI, Redlands, USA). As dolphins are top-order predators it is assumed their distribution is highly influenced by prey availability (Eierman & Connor, 2014). Prey abundance information for Port Phillip Bay is limited and, therefore, environmental variables known to influence their distribution were selected as proxies for habitat suitability models. Dynamic environmental variables such as sea surface temperature and sea surface chlorophyll-a are available only at low spatial resolution across Port Phillip Bay (SST: 0.05° × 0.05° (Good et al., 2020); Chl-a: 0.042° × 0.042° (Sathyendranath et al., 2019)). Although they vary seasonally, the low resolution and variation suggested that these variables were not useful in the subsequent analyses. Instead, static environmental variables that have been shown to influence the distribution of bottlenose dolphins in previous studies (Beddoe et al., 2024; Hanf et al., 2022; Haughey et al., 2021) were incorporated into habitat suitability models. Temperature, light attenuation and the presence of benthic primary producers are influenced by seafloor depth, and areas of upwelling that attract fish and cephalopods are reflected through associated seafloor slope (Eierman & Connor, 2014). Therefore, bathymetry (m) and seafloor slope (% gradient) were included as predictor variables in the models. Bathymetry data for Port Phillip Bay were sourced from a Coastal Digital Elevation Model created by the Cooperative Research Centre for Spatial Information (CRCSI) (2022) with a resolution of 10 m × 10 m and depth values ranging from 0 m (mean sea level) to −80 m. The benthic terrain modeller in ArcGIS was used to calculate slope, aspect and TPI from bathymetry (Walbridge et al., 2018). To obtain perspective of fine-scale gradients, slope was calculated using a focal window of 10 m × 10 m.

Physical topographic features of the seafloor, such as its ruggedness and aspect, can influence ocean currents and tides that effect prey distribution (Bailey & Thompson, 2010). Therefore, seafloor aspect (degrees) and Topographic Position Index (TPI) were additionally included as predictor variables in the models. TPI is a measure that reflects upper, middle and lower features of the landscape, with an index of 0 or near 0 reflecting flat or continuous features, a large positive value reflecting ridges or mounts, and a large negative value reflecting the bottom of a ridge or shelf (Amatulli et al., 2018). The sine and cosine of aspect, representing the relative eastness and northness of downslope direction (Hession & Moore, 2011), were calculated to prevent problematic logistic models. As features of the study area are small-scaled due to the relatively flat landscape of Port Phillip Bay (Holdgate et al., 2001), TPI was calculated based on a cell neighborhood of 5 × 5 grid cells each 10 m × 10 m.

Seafloor habitat type (i.e. sediment, seagrass beds, macroalgae on sediment, biogenic reef, rocky reef and sessile invertebrate beds) directly influences the presence of a diverse range of invertebrate, fish and cephalopod species (Eierman & Connor, 2014). Therefore, benthic habitat was included as a categorical predictor variable in the model. Benthic habitat data were sourced from a Statewide Marine Habitat Map (2023) (10 m × 10 m spatial resolution), within the DEECA CoastKit database (Mazor et al., 2023).

Human activities occurring in Port Phillip Bay have been suggested to influence bottlenose dolphin behavior and habitat relationships (Filby et al., 2017; Scarpaci, Corkeron & Nugegoda, 2003). Hence, two variables reflecting the environment in relation to urbanization and vessel (commercial and recreational) activity were included as predictor variables in the models as Euclidean distance to coastline (m) and Euclidean distance to shipping channels (m), respectively. As previous studies have found these factors to reduce the likelihood of bottlenose dolphin presence in habitat areas (Filby et al., 2017; Zanardo et al., 2017), it is hypothesized that probability of presence will increase with increasing distance (m). These two predictor variables were calculated using geospatial vectors and spatial analyst in ArcGIS.

Raster layers were generated for the extent of Port Phillip Bay with grid cell resolution of 10 m × 10 m. This resolution was chosen to ensure fine-scale sensitivity when analyzing bathymetric features and Euclidean distances.

Ensemble habitat suitability modelling

Statistical modelling of habitat associations with dolphin distribution was conducted in R statistical environment (version 4.4.0) (R Core Team, 2024). An ensemble presence-background habitat suitability model was used to predict potential habitat areas in Port Phillip Bay most suitable for bottlenose dolphins and to assess the importance of environmental variables.

As true/inferred absences and search effort information were lacking in the present study random background points were generated. To reduce the effect of sampling bias, background points were extracted within the model calibration zone (area sampled by ecotourism and other vessels of opportunity). Environmental variables were clipped at a 2 km radius of presence points and 3,000 random background points were sampled within this radius. The model calibration zone was defined considering visibility range relative of pseudo-absence and height of the observation platform. Three background point sample replicates were generated to account for variation within each modelling algorithm. If multiple presence or background points were recorded in a single 10 m grid cell, these were combined to represent a single presence or background point to avoid pseudo-replication in the model (Correia et al., 2021; Elith et al., 2011). Once having removed NA (not available) values from the presence-background data, statistical assumptions could be checked and met.

Before running habitat suitability models, collinearity (high correlation) between predictor variables was assessed using ‘cor’ from the car package (Fox & Weisberg, 2019), and ‘vif’ from the usdm package (Naimi et al., 2014). Variance inflation factors (VIF) and correlations between continuous variables were computed to allow the exclusion of highly correlated variables from the same models, using a linear correlation threshold of 0.7 and VIF threshold of 3 (Haughey et al., 2021). As VIFs and coefficients were below the thresholds for the dataset, all seven predictor variables were included in the same models.

An ensemble habitat suitability model was generated using the BioMod2 package (Thuiller et al., 2009). Predictions from five different presence-background modelling algorithms; flexible discriminant analysis (FDA), generalized additive model (GAM), generalized boosted model (GBM), multivariate adaptive regression splines (MARS) and maximum entropy (MAXENT), were combined. These algorithms were selected as they perform fairly well in their predictive accuracy (Phillips, Dudík & Schapire, 2004; Thuiller et al., 2009) and allow an overall comparison between regression (GAM, GBM, MARS), classification (FDA) and machine learning (MAXENT).

Default algorithm parameters were used with each FDA, GAM, GBM and MARS, as recommended by Thuiller et al. (2009). However, MAXENT parameters were refined using ‘ENM evaluate’ from the ENMeval package (Muscarella et al., 2014), with preliminary analyses having indicated linear and quadratic features to provide the greatest area under the receiver operating characteristic curve (AUC) score and smallest ΔAICc. Model algorithm settings are listed in Material S2. Habitat suitability models were built with a binomial error distribution and ‘logit’ link function. Data was randomly split into model calibration (75%) and evaluation (25%) data, following a 10-fold cross-validation method (Thuiller et al., 2009). To prevent negative prevalence effects, presence and background points were equally weighted with a set prevalence of 0.5.

The AUC metric was used to assess model predictive performance. An AUC score < 0.5 suggests that a model performs worse than random (i.e. counter-predictions). Model performance was determined following (Araújo et al. 2005): AUC < 0.5 = counter-predictions; 0.5 < AUC < 0.6 = fail; 0.6 < AUC < 0.7 = poor; 0.7 < AUC < 0.8 = fair; 0.8 < AUC < 0.9 = good; 0.9 < AUC = excellent. Only algorithms with a mean AUC value > 0.7 were included in further analyses to ensure sufficient prediction accuracy. The importance of environmental variables was determined using a randomized 10-permutation run procedure within BioMod2 (Thuiller et al., 2009). This approach calculates the Pearson’s correlation between standard predictions using all predictor variables and predictions where the variable being evaluated is randomly permutated (rearranged). Low correlation (i.e. significant difference between the two predictions of variable importance) indicates the predictor variable is of high importance, and high correlation indicates the variable is of low importance. According to the mean correlation coefficient, variables are then ranked from 0 to 1, with the highest-ranking variable the most influential and the lowest the least influential (Thuiller et al., 2009).

To generate the final ensemble model, the mean AUC score of all algorithms if >0.7 was calculated. The importance of each environmental variable across the different model algorithms was averaged using a mean of means value. An ensemble habitat suitability map illustrating mean predicted probability and associated variance from model algorithms was generated within the model projection zone. Response curves describing the variation in probability of bottlenose dolphin presence as a response of each environmental variable were plotted.

Results

A total of 523 bottlenose dolphin sightings were obtained from spring to autumn during 2006–2023, from which 392 (75%) were from patrons and crew onboard ecotourism vessels and 131 (25%) from opportunistic vessels (VBA) (Table 1). The number of collected sightings varied greatly between years with the most sightings having been from three ecotourism companies during 2022 (n = 118). No VBA or ecotourism data was available for years 2018 and 2020 due to extreme weather events and the global coronavirus pandemic.

Table 1 Summary of bottlenose dolphin sightings obtained each year from various data sources within Port Phillip Bay.

Data missing for years 2018 and 2020.

Year	Unique presences (n)	Data source	
2006	15	VBA	
2007	25	VBA	
2008	7	VBA	
2009	15	VBA	
2010	14	Sea All Dolphin Swims, VBA	
2011	75	Sea All Dolphin Swims, VBA	
2012	80	Sea All Dolphin Swims, VBA	
2013	59	Sea All Dolphin Swims, VBA	
2014	8	Polperro Dolphin Swims, VBA	
2015	29	Polperro Dolphin Swims, VBA	
2016	26	VBA	
2017	2	VBA	
2019	11	Moonraker Dolphin Swims, Polperro Dolphin Swims	
2021	2	Moonraker Dolphin Swims	
2022	118	Sea All Dolphin Swims, Moonraker Dolphin Swims,
Polperro Dolphin Swims,	
2023	37	Sea All Dolphin Swims, Moonraker Dolphin Swims	
Total unique presences (n)	523		

Dolphin sightings used to build habitat suitability models were largely restricted to coastal waters of Port Phillip Bay (Fig. 1B), occurring almost exclusively (97.4%) in areas with depths <20 m. Nonetheless, all individual model algorithms had a fairly good performance (AUC range = 0.74–0.9, median = 0.8) (Fig. 2) and were therefore included in the ensemble model. Apart from MARS and GBM algorithms individual model performance was equal to or less than the ensemble model, which had an AUC score of 0.81 indicating good predictive performance (Fig. 2). The GBM algorithm generated the highest AUC score overall (0.89).

Figure 2 Performance of habitat suitability models.

Boxplots presenting AUC score range of the 10-cross validation runs of each modelling algorithm and presence-background sample within. The red line indicates the AUC score of the ensemble model.

The ensemble model indicated distance to shipping channels (m) and distance to coastline (m) as the two most influential variables (0.68), having accounted for the most variation in dolphin occurrence (Table 2). In contrast, seafloor aspect and TPI were of least influence (0.016) and therefore accounted for the least amount of variation overall. All five individual model algorithms agreed that distance to shipping channels (m) had the greatest influence on dolphin distribution (Table 2).

Table 2 Importance of environmental variables used in habitat suitability models.

Model algorithm	Environmental variable	
	Bathymetry (m)	Seafloor slope (% gradient)	Seafloor aspect NS (cosine)	Seafloor aspect EW (sine)	TPI	Benthic habitat	Distance to shipping channel (m)	Distance to coastline (m)	
FDA30	0.08	0.05	0.003	0.001	0.006	0.16	0.47	0.41	
GAM30	0.06	0.06	0.003	0.001	0.006	0.11	0.49	0.40	
GBM30	0.14	0.19	0.009	0.002	0.02	0.01	0.36	0.19	
MARS30	0.23	0.17	0.004	0.004	0.02	0.07	0.29	0.14	
MAXENT30	0.30	0.06	0.002	0.003	0.003	0.15	0.38	0.27	
Mean of means/Ensemble	0.16	0.11	0.004	0.002	0.01	0.10	0.40	0.28	
Note:

Variable importance is presented as the mean value over the 30 model runs of each algorithm, and the mean of means amongst them/ensemble value. Displayed in subscript is the number of runs of each algorithm (with an AUC evaluation score > 0.7) included in the ensemble model. Highlighted in bold are the variables of greatest importance.

Response curves present negative relationships between the likelihood of dolphin occurrence, and distance to shipping channels (m) and coastline (m), with probability decreasing with increasing distance (Fig. 3). This indicates that bottlenose dolphins in Port Phillip Bay were more likely to be present in waters either closer to shipping channels or the coastline.

Figure 3 Response curves of the five modelling algorithms used in the ensemble.

Modelling probability of bottlenose dolphin occurrence within Port Phillip Bay as a response to bathymetry (m), seafloor slope (% gradient), seafloor aspect NS and EW, TPI, benthic habitat, distance to shipping channels (m) and distance to coastline (m). Benthic habitat categories are as follows: 1-sediment; 2-seagrass beds; 3-macroalgae on sediment; 4-biogenic reef; 5-rocky reef; 6-sessile invertebrate beds.

Bathymetry (m) and associated seafloor slope (% gradient) which combined had relatively high importance (>0.25) in the ensemble model (Table 2), were shown to have non-linear relationships with the likelihood of dolphin occurrence for MARS, GBM and MAXENT (Fig. 3). Probability of presence peaked at depths ~12 m and slopes with a 10–15% gradient. This indicates that bottlenose dolphins in Port Phillip Bay were more likely to occur in relatively shallow waters where seafloor slopes are prominent. In comparison, the influence of benthic habitat type on dolphin presence was relatively minor in the ensemble model (0.10). Nonetheless, a slight preference for seagrass beds and biogenic reef was evident (Fig. 3). Seafloor aspect displayed minimal influence on the probability of dolphin presence (Table 2).

Habitat suitability predictions from the ensemble model indicated clear distinctions between highly suitable and unsuitable habitat (Fig. 4A). The predicted areas of high suitability were in coastal waters close to Hobsons Bay (Melbourne), Sorrento, Queenscliff and Corio Bay/Geelong. In contrast, deeper waters in the center, and shallow coastal waters in Werribee South Foreshore and the eastern extent, of Port Phillip Bay were predicted to be of lowest habitat suitability. Error in the uncertainty of predictions was relatively low, apart from southern channel areas near Port Phillip Heads and Sorrento (Fig. 4B).

Figure 4 Habitat suitability predictions.

(A) Ensemble habitat suitability predictions and (B) variance for bottlenose dolphins within Port Phillip Bay at a 10 m × 10 m cell resolution (map source: Victorian Department of Energy Environment and Climate Action (2024)). Habitat suitability is highest at ‘0.99’ and lowest at ‘0’.

Discussion

Understanding the influence of environmental factors on species distribution, ecology and population abundance is required to ensure robust conservation management and species long-term viability (Charbonnel et al., 2022; Moxley et al., 2017; Zotos, Stamatiou & Vogiatzakis, 2022). Obtaining such information for elusive marine mammals is financially and logistically challenging and alternative economically viable methods of data collection are needed (Kaschner et al., 2012; Redfern et al., 2006). In the present study, ecotourism vessels in southern Port Phillip Bay, south-eastern Australia, were used as platforms to collect location information on a poorly studied bottlenose dolphin population. These data were used to build an ensemble habitat suitability model, from which static environmental factors known to influence dolphin distribution were investigated, and important habitats and potential distribution identified. The model suggested the most suitable habitats within Port Phillip Bay were those in close proximity to shipping channels and the coastline. The findings highlight the importance of community-based data collection and have implications for the management of this small dolphin population.

While a recent preliminary study investigated bottlenose dolphin habitat use in Port Phillip Bay (Beddoe et al., 2024), its extent was limited relative to available distribution/habitat and did not provide predictive information. Bottlenose dolphin sighting hotspots during austral summer and spring did not coincide with those areas of highest habitat suitability in the present study. However, sighting hotspots during the austral autumn period alone did coincide with important habitat areas. The findings from this preliminary study highlight the importance of habitat suitability predictions to determine the species potential distribution in Port Phillip Bay, essential for its effective conservation and management.

Habitat suitability model implications

While the ensemble model had a good predictive performance, identifying important habitats for bottlenose dolphins in Port Phillip Bay, three sources of bias may have influenced the results. As the data used to generate the model were collected within the operational limits of ecotourism boats and other vessels of opportunity, dolphin sightings were largely restricted to coastal waters. Sightings were scarce within central Port Phillip Bay where depth is greatest and human activity is minimal (Holdgate et al., 2001). Firstly, presence sampling bias is a common issue for presence-background models, as sightings used are generally from easily accessible platforms of opportunity (Phillips et al., 2009). While in the present study data were not collected equally across the predicted habitat areas, previous studies have shown long-term opportunistic data collection is a cost-effective method for identifying distribution trends for elusive marine and terrestrial species (Currie et al., 2018; Matutini et al., 2021). Nonetheless, additional data from central Port Phillip Bay is required to further validate the model predictions derived in the present study. Secondly, repetitive sampling of individuals is a common bias associated with opportunistic data collection from ecotourism operators, as the same individuals are often revisited on different trips within the same day (Filby, Stockin & Scarpaci, 2014; Lacetera et al., 2023). While in the present study ~16% of all records were within a distance of 500 m, further data processing is required to ensure dolphin sightings are distant enough to assume they are not the same individual. Indeed, bottlenose dolphins have been observed traveling ~30–50 km per day within relatively shallow, tidal dominated embayment’s (Bearzi, Bonizzoni & Gonzalvo, 2011; Wilson, 2022), and therefore, it may be unlikely an individual or pod would remain for extended time periods in areas sampled by ecotourism operators. Thirdly, the use of pseudo-absences instead of true absences is a universal bias associated with presence-background models, as random samples can include a location where the species may occur but was not detected. In the present study, this source of bias was reduced using a radius-restricted background, which has similarly been applied in studies predicting habitat suitability for common bottlenose dolphins in the north-Atlantic (Correia et al., 2021). As model predictions reflect the species fundamental niche and measure the suitability of habitat features similar to where dolphin sightings did and did not occur, caution was taken when interpreting model outputs.

While presence-absence data is ideal for habitat suitability modelling (provides transects with presences and inferred absences) it requires more intensive survey effort and is usually collected systematically using research vessels (Fiedler et al., 2018; Zanardo et al., 2017). When comparing presence-absence and presence-background habitat suitability models, studies have found that high-quality inferred absence data greatly improves model performance and accuracy of predictions (Bradter et al., 2021). For example, systematic survey data (presence-absence) for sperm whales (Physeter macrocephalus) in the northwest Mediterranean Sea generated higher performing habitat suitability models compared to opportunistic presence-background data (Praca et al., 2009). Systematic surveys in Port Phillip Bay are not logistically or financially feasible, even with data collected opportunistically by ecotourism vessels due to their current operational limits. Thus, presence-background modelling may be a more useful approach for vessels of opportunity data collection. Indeed, previous research modelling habitat suitability for four whale species in the eastern tropical Pacific found no significant difference in model performance generated with opportunistically collected citizen science data (presence-background) (AUC = 0.78 ± 0.14) and systematic surveys (presence-absence) (AUC = 0.79 ± 0.15) (Fiedler et al., 2018). This implies that habitat suitability models from community-based data collection may have important implications for future conservation management of bottlenose dolphin populations.

Influence of environmental factors

The results of the present study suggest that areas close to shipping channels and coastlines are important habitats for bottlenose dolphins in Port Phillip Bay. These features are likely to be associated with favorable foraging conditions for dolphins (Bailey & Thompson, 2010; Eierman & Connor, 2014), and are similar to those influencing dolphin distribution in other studies (Haughey et al., 2021). For example, distance from coastline had the greatest influence on common bottlenose dolphin distribution along the Liguria coast (north-western Mediterranean Sea), with individuals preferring areas near shallow coastal waters (Marini et al., 2015). Similarly, in waters of Osa Peninsula, Costa Rica, coastal seafloor slope was found to have a considerable effect on the distribution of common bottlenose dolphins, with individuals preferring steep habitats closer to shore (Pacheco-Polanco et al., 2019).

Dredged shipping channels are associated with steep seafloor slopes which create diverse vertical habitats that are known to be associated with higher densities of fish and cephalopods (Smith & Lindholm, 2016). In addition, cetaceans are known to use steep seafloor slopes to concentrate prey into shallow waters where they may be easier to capture (Eierman & Connor, 2014; Sargeant & Mann, 2009). This is a foraging strategy suggested for common bottlenose dolphins in the Moray Firth Scotland, with previous research having found steep seafloor slopes to be a preferred foraging habitat for the species (Bailey & Thompson, 2010).

While the movement and presence of container and cargo ships, fishing vessels and recreational boats can deter dolphins from suitable habitat (Filby et al., 2017; Marley et al., 2017), the results of the present study suggest that this may not be the case for bottlenose dolphins in Port Phillip Bay. The bays steep southern shipping channels frequently exchange tidal flow with Bass Strait influencing nutrient and prey availability (Harris et al., 1996; Sampson, Easton & Singh, 2014). Similarly, the western channels are associated with steep coastal gradients and dense seagrass habitat which may promote more efficient prey capture (Eierman & Connor, 2014; Mazor et al., 2023). In addition, dolphins often engage in bow-riding as a form of social interaction or when travelling between locations (Filby, Stockin & Scarpaci, 2014; Fish, Gough & Adams, 2024). By riding the bow wave of vessels dolphins can efficiently travel greater distances and conserve energy for foraging and reproduction (Fish, Gough & Adams, 2024).

Seagrass beds are commonly found in shallow coastal waters of Port Phillip Bay and are known to provide ideal breeding and spawning conditions for fish and cephalopod species such as southern garfish (Hyporhamphus melanochir) and southern calamari (Sepioteuthis australis) (Moltschaniwskyj & Pecl, 2003; Nuttall, Stewart & Hughes, 2012), common prey items for bottlenose dolphins (Zanardo et al., 2017). Previous research in Gulf St Vincent, South Australia, has also revealed the distribution of a resident bottlenose dolphin population to be associated with benthic habitats consisting of seagrass beds (Zanardo et al., 2017).

Acknowledging the deeper central areas of Port Phillip Bay were beyond the model calibration zone and require further sampling, the results of the present study indicated that these were not of high habitat suitability. This contrasts with deep (>20 m) offshore waters of the Exmouth Gulf, Western Australia, being important habitat of Indo-Pacific bottlenose dolphins (Hanf et al., 2022). Similarly, the deep (>200 m) waters of the central Gulf of Mexico have suitable habitat for four different dolphin species (Ramírez-León et al., 2021). Deep waters are commonly associated with steep sloping features such as canyons, ridges and seamounts that are nutrient rich and attract an abundant supply of schooling fish and cephalopod prey species (Borland et al., 2021; Smith & Lindholm, 2016). However, the deeper central area of Port Phillip Bay is characterized by relatively flat topography and muddy sediments, and is predominantly inhabited by benthic sessile invertebrates (Holdgate et al., 2001; Mazor et al., 2023). Such species are not commonly consumed by bottlenose dolphins (Paschoalini & Santos, 2020) and, therefore, central Port Phillip Bay may not be a profitable foraging zone.

In the present study topographic seafloor features (i.e., seafloor aspect and TPI) did not greatly influence the distribution of bottlenose dolphins in Port Phillip Bay. This is in contrast to common dolphins (Delphinus delphis) in the coastal waters of northwest Spain, where physical seafloor features strongly influence the species’ distribution (Paradell, López & Methion, 2019). In areas where steep slopes occur, topographic seafloor features may have a greater influence on dynamic factors such as ocean currents, upwellings and temperatures which in turn can have considerable effects on prey distribution (Bailey & Thompson, 2010; Hanf et al., 2022; Ramírez-León et al., 2021). As Port Phillip Bay is relatively shallow and enclosed, topographic features such as seafloor aspect and TPI may not greatly influence current flow or changing temperature and, thus, prey distribution (Harris et al., 1996; Holdgate et al., 2001).

Previous studies have shown dynamic environmental parameters can greatly influence the distribution of cetaceans (Paradell, López & Methion, 2019). For example, in the coastal waters of western and southern Portugal, sea surface chlorophyll-a concentration had the greatest influence on the distribution of common dolphins, with the species preferring waters with higher concentrations (Moura, Sillero & Rodrigues, 2012). Similarly, the distribution of Indo-Pacific bottlenose dolphins in coastal waters of Western Australia was predominantly influenced by sea surface temperature during austral winter with the species preferring water temperatures of ~23.5° (Haughey et al., 2021). Correspondingly, tide phase largely influenced the distribution of two cetacean species in waters of a semi-enclosed embayment near the Irish Sea, with species more likely to occur within the bay during low (ebb) tides (de Boer et al., 2014). Port Phillip Bay being a semi-enclosed embayment is similarly characterized by tide cycles that may influence prey availability and distribution and can restrict dolphins foraging in shallow habitats at certain time periods (Fury & Harrison, 2011; Harris et al., 1996). To explore the effects of dynamic parameters on dolphin distribution, high quality remotely sensed data is required. However, sea surface temperature and sea surface chlorophyll-a data are not accurate at small spatial scales, especially in shallow regions (Fernández-Tejedor, Velasco & Angelats, 2022; Smit et al., 2013) such as Port Phillip Bay. Similarly, tide phase data currently lacks spatial coverage in the bay and is required across a greater extent to address the effects of tide on dolphin distribution.

Conclusions

In summary, the present study has revealed regions with steep seafloor slopes or coastal seagrass beds to be important habitat types for bottlenose dolphins in Port Phillip Bay. The study has also highlighted the value of presence-only data from ecotourism operators and vessels of opportunity for providing information on critical habitat needs for the species. However, as dolphin sightings used to generate the model were primarily restricted to coastal waters of Port Phillip Bay, sampling bias may have influenced model results. Therefore, for future studies, opportunistic data from additional vessels (e.g., passenger ferries, pilot boats, cruise ships, fishing charters and cargo/container vessels) that cover a greater sampling area of Port Phillip Bay is required, particularly from the center of the bay where bottlenose dolphin sightings are extremely limited. Consequently, this emphasizes the benefit of collaborations with ecotourism operators and their patrons for the continued collection of such critical information for this and other bottlenose dolphin populations.

Seasonal variation in habitat use has recently been documented for bottlenose dolphins in Port Phillip Bay (Beddoe et al., 2024). This presents a unique opportunity to determine the influence of local anthropogenic factors (i.e. extreme sea surface temperatures and nutrient levels) on the abundance and distribution of prey populations important to bottlenose dolphins (Fury & Harrison, 2011; Paradell, López & Methion, 2019). However, additional sampling within Port Phillip Bay throughout different times of the year may be required to better understand these habitat influences. In addition, genetically similar bottlenose dolphins have been shown to inhabit the Gippsland Lakes (Charlton-Robb, Taylor & McKechnie, 2015) and are known to other areas such as the northern Tasmanian coastline (Möller et al., 2008). As these regions may have different habitat features, sampling should be extended to them in order to develop more widely applicable habitat suitability models. Such information is necessary for determining the critical habitat needs of, and, thus, to better manage, these vulnerable bottlenose dolphin populations. The use of ecotourism operations, and other vessels of opportunity, may provide the low-cost, high data volume collection regime needed for achieving these goals.

Supplemental Information

Supplemental Information 1 Geo-referenced locations of bottlenose dolphin sightings used for habitat suitability models.

Sightings sourced from the Victorian Biodiversity Atlas (VBA) database and 3 ecotour vessels (Sea All Dolphin Swims, Moonraker Dolphin Swims and Polperro Dolphin Swims) in southern Port Phillip Bay.

Supplemental Information 2 BioMod2 algorithm settings.

Supplemental Information 3 Environmental variable raster layers used to generate habitat suitability models for Port Phillip Bay.

Variables include: (a) bathymetry (m), (b) seafloor slope (% gradient), (c) seafloor aspect NS and (d) EW, (e) Topographic Position Index (TPI), (f) benthic habitat, (g) Euclidean distance to shipping channels (m) and (h) coastline (m) (map sources: CRCSI (2022); DEECA (2023); Mazor et al. (2023)). Benthic habitat categories are as follows: 1 - sediment; 2 - seagrass beds; 3 - macroalgae on sediment; 4 - biogenic reef; 5 - rocky reef; 6 - sessile invertebrate beds.

We thank the patrons, staff and volunteers of the Sea All Dolphin Swims, the Moonraker Dolphin Swims and the Polperro Dolphin Swims for their efforts in collecting the data for this project. We especially thank James Murphy for his contribution to this project.

Additional Information and Declarations

Competing Interests

Author Contributions

Data Availability

The authors declare that they have no competing interests.

Maddison J. Ledwidge conceived and designed the experiments, performed the experiments, analyzed the data, prepared figures and/or tables, authored or reviewed drafts of the article, and approved the final draft.

Jacquomo Monk conceived and designed the experiments, analyzed the data, authored or reviewed drafts of the article, and approved the final draft.

Suzanne J. Mason conceived and designed the experiments, authored or reviewed drafts of the article, and approved the final draft.

John P. Y. Arnould conceived and designed the experiments, analyzed the data, authored or reviewed drafts of the article, and approved the final draft.

The following information was supplied regarding data availability:

The geo-referenced locations of bottlenose dolphin sightings used for the habitat suitability model is available in the Supplemental File.

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
