# Peer review of "Using vessels of opportunity for determining important habitats of bottlenose dolphins in Port Phillip Bay, south-eastern Australia"

_PeerJ, doi:10.7717/peerj.18400_

## Round 0.1 · original submission · Major Revisions

I have received two independent reviews of your study. While both reviewers clearly recognised the quality/novelty of your work, they have collectively raised a number of major issues that will need to be addressed in your revised manuscript.

Particularly, it was noted that the current manuscript lacks some important details throughout the manuscript and that clarification should be provided to justify the methodology used and to support the conclusions with more convincing arguments.

Overall, the reviewers have provided you with excellent suggestions on how to improve the manuscript, and I be looking forward to receiving your revised manuscript along with a point-by-point response to their comments.

With warm regards,
Xavier

Reviewer 1 ·

Basic reporting

In this study, the authors aim to discover patterns of habitat suitability for bottlenose dolphins occurring in Port Phillip Bay (southeastern Australia) through the use of visual data from platforms of opportunity. Besides providing a good overview regarding the importance of using modeling approaches for conservation and management, the authors have conducted presence-only models to estimate habitat suitability in Port Phillip Bay for bottlenose dolphins, allowing a strong comprehension of the environmental/anthropogenic factors affecting dolphin distribution. These results have important repercussions regarding potential conservation measures to protect this under-studied population. Therefore, taking into account the conservation status of this population, together with the scarcity of primary information, I do consider that the study is relevant and worthy of being published in an international journal.

The article is well-written and pleasant to read. I particularly appreciated the inclusion of environmental variables (e.g., ruggedness and aspect) usually not frequently used in environmental niche modeling. However, some specific issues are weakening the study, in my point of view. They will be detailed in the following sections.

Experimental design

In general, the methods are clear and replicable. However, when reading this study, I have had some major concerns regarding the methodology employed by the authors.
. [lines 151-153] The authors explained their approach to avoiding duplicated sightings, essential to not introduce more bias in the predictions. In my experience with whale-watching companies, boats usually revisit the same animals on different trips in the same day. This approach seems insufficient, and there may still be duplicated records in your data. I recommend considering sightings on the same day that are distant enough to assume they are not the same animals (authors should consult relevant bibliographic references on distances traveled by bottlenose dolphins in a day). Alternatively, this bias should be highlighted, and its implications explained in the Discussion section when referring to the limitations of this study. Additionally, I noticed that no sighting times are provided in Supplementary File 1.
. [lines 178-179] The authors stated that they found only data for sea surface temperature and chlorophyll at coarse resolution. I recommend the authors thoroughly search for this kind of data, as its use is crucial for the overall predictions of the models. Alternatively, consider the use of these two environmental variables even if at a lower resolution. How is the performance of the models when including these two variables?
. [lines 236-237] To reduce the effect of sampling bias, background points should be located in areas whose habitat features are in the same proportion as the location of the sightings. I did not understand whether the authors had done that. Please clarify.

Validity of the findings

While the findings obtained are indisputable, and the authors have thoroughly discussed them, I am particularly concerned about the extent to which the authors have drawn their conclusions. As the authors correctly pointed out, the use of presence-only models presents several limitations. In this case, the models considered measure the suitability in habitats whose features are similar to those locations where dolphins are found. This should be emphasized in the Discussion section. I would also advise being careful when drawing conclusions.
- [lines 287-288] I was triggered by the negative relationship between the likelihood of dolphin occurrence and distance to shipping channels. While the authors have provided a possible explanation for this pattern, I was wondering how a possible increase in the number of vessels may have not “discouraged” dolphins to leave those areas. I think this is a point worth to discuss, even if briefly.
- [lines 408-411] Acknowledging the presence of these oceanographic features emphasizes the need to include dynamic environmental variables (sea surface temperature, chlorophyll, sea level anomaly, etc) in your models for more accurate predictions.

Additional comments

General comments regarding the manuscript:
. In the Abstract (lines 15-16), it should read “Understanding species' critical habitat requirements is crucial for effective conservation and management.”
. [Lines 26-28] From the Abstract, the authors state that the high performance of Maxent models means that these models are accurate enough to predict species distribution. I do not agree with this statement. Maxent models might (should) be used when limited data is available; however, one must be cautious when drawing robust conclusions from these approaches. For example, in cases where data is not collected equally across the predicted habitat areas (as in this study), we can only infer part of the habitat suitable for a specific species. I recommend rephrasing this sentence.
In the Introduction section:
. [Lines 53, 62, and 323] I would use “elusive” instead of “cryptic.”
. [Lines 86-87] The authors should change the current number of Tursiops species. According to the Committee on Taxonomy from the Society of Marine Mammalogy, there are currently 3 Tursiops species: Tursiops truncatus, Tursiops aduncus, and Tursiops erebennus.
. [Line 103] The authors can mention Figure 1.
In the Material and Methods section:
. When characterizing the study area, the authors should also provide information regarding other types of vessels present in the area (cargo ships, cruise ships, fishing boats, etc.). Try to provide numbers so the reader will understand the real impact of these activities on this dolphin population.
. [Lines 226-228] The authors assessed multicollinearity between variables using Pearson correlation. I strongly recommend the authors also assess multicollinearity through Variance Inflation Factor (VIF).
In the Discussion section:
. [Line 354] The authors refer to presence-absence data. I would like to emphasize that in cetacean research, there is never absence data (even when we know the survey effort and the transects performed to collect the data). It is always pseudo-absence data.
. [Lines 357-359] The authors state that performing systematic surveys in Port Phillip Bay is not logistically or financially feasible. I have no doubts about that. Nevertheless, I think the authors could suggest the use of other platforms of opportunity operating in the bay (rather than ecotour vessels) to have a higher monitoring effort and consequently have data in a higher temporal and spatial frame.
. [Lines 382-383] Have the authors observed or found reported any similar foraging strategy in dolphins of Port Phillip Bay?

·

Basic reporting

The article does not always use unambiguous and technically correct text.
A few important details are missing from a paragraph in the introduction.
There are too many references (sometimes there are more than three references cited to support the same argument) and there are several old references that could be replaced by more recent references that are already cited elsewhere.
The different information from current figures 1 and 2 should be merged into a single figure (new future figure 1) by adding some details to this new figure.

Experimental design

The investigation was not entirely conducted rigorously and to a high technical standard.
Methods were not entirely described with sufficient and clear information.

Validity of the findings

no comment

Additional comments

# GENERAL COMMENTS
The original research question was well defined, relevant and meaningful. It is stated how the research fills an identified knowledge gap: the suitable habitats of a small poorly-studied population of bottlenose dolphins in Port Phillip Bay (south-eastern Australia) that has recently been proposed to comprise a unique newly described species that is listed as ‘Critically Endangered’, the Burrunan dolphin (Tursiops australis). This study is therefore important for the effectiveness of conservation management of this species. The English language and grammar rules are used correctly.

However, there are some important issues, particularly in the choices made for modelling, which must be treated carefully and by providing both important details and convincing arguments that are missing at the moment before any potential acceptance for publication. These important issues are:
- The paragraph on lines 102-111 of the introduction should be completely revised so that it is clearer with regard to the possible interactions and or potential spatio-temporal overlaps between the 3 species (T. truncatus, T. aduncus and T. australis) and also between the two populations of T. australis (the one in the Port Phillip Bay and the other in the Gippsland Lakes) while re-mentioning the term “cryptic species”.
- The sub-section “Environmental variables” on lines 171-216 should be modified by moving the elements coming from the last paragraph into the other paragraphs in order to have a more fluid reading by passing from one variable to another without going back again on a variable already mentioned before. This subsection should be more complete by adding i. some missing information regarding the spatial resolution and/or the source of certain environmental variables, and ii. underlying assumptions of what you might expect before getting your results for each of the seven environmental variables used in the model in order to ensure rigor of your scientific reasoning. You will read my suggestions in more detail below.
- There are missing important details and convincing arguments in the sub-section “Habitat Suitability Modelling” on lines 218-261:
-- In recent years, we have entered the era of ensemble forecasting (or modelling) (Lasram et al. 2020, Schickele et al. 2020, Charbonnel et al. 2022, Navarro et al. 2023 and other references). Indeed, as recommended in the literature, ensemble forecasting was used instead of strict selection of a single statistical technique was used for habitat suitability distribution models because ensemble forecasting allows the variations in the accuracy of predictions produced from several statistical techniques (Araújo and New, 2007, Marmion et al. 2009; Hao et al. 2019) such GLM, GAM, MARS, CTA, FDA, ANN, RF, GBM and MaxEnt from the R package biomod2 (Thuiller et al. 2009). Moreover, statistical techniques other than MaxEnt also allow the inclusion of background points as explained, for instance in Lasram et al. 2020, Schickele et al. 2020, Charbonnel et al. 2022 and Navarro et al. 2023. Indeed, regression techniques based on presence and absence data have been showed to work better than presence-only techniques, stratified pseudo-absence data (i.e., background data) should be generated (Brotons et al. 2004). I would like to point out to you that you have used this regression technique (presence points + background points) and not the presence-only technique. This term “presence-only” should therefore be corrected throughout the manuscript in order to be more precise in terms of the vocabulary associated with habitat suitability distribution modelling.
It is therefore not possible to ignore this ensemble forecasting with biomod2 in your article and to use a single algorithm without providing convincing arguments in order to demonstrate the adequacy of your methodological choice using only a single algorithm (MaxEnt). I recommend that you also consult Guisan et al. (2017).
-- Details are missing regarding the selection of background points and the prevalence of your model. Were these background points extracted only in the area corresponding to the model calibration zone (which is the zone sampled by ecotourism and other vessels of opportunity) or within the projection zone (which is the entire area covering the entire Port Phillip Bay)? Indeed, to avoid any bias linked to the absence of sampling in certain areas of the Port Phillip Bay, the background points must be selected only in the areas which were sampled by ecotourism and other vessels of opportunity. Please read up on this, for example in Lasram et al. (2020), Schickele et al. (2020), Charbonnel et al. (2022), Navarro et al. (2023) and other recent references, to learn more about the rigor of their background (or pseudo-absence) points selection. Moreover, you used n = 534 presence points and 5,000 random background points without providing verification details regarding any bias linked to negative prevalence effects.
-- I highly recommend you read the fairly recent book "Habitat suitability and distribution models with applications in R" written by A. Guisan, W. Thuiller and N.E. Zimmermann (2017) and other recent references in distribution modelling because you wrote errors regarding model evaluation and variable importance. You will read my suggestions in more detail below.
- The different information from current figures 1 and 2 should be merged into a single figure (new future figure 1). You will read my suggestions in more detail below.
- There are too many references (sometimes there are more than three references cited to support the same argument) and there are several old references that could be replaced by more recent references that are already cited elsewhere. Most beginners make these faults in writing scientific articles. You will read my suggestions in more detail below.
- It would be desirable to add the mapping of values/categories for each of the environmental variables used in the habitat suitability modelling across the entire study area in supplemental files in order to visualize the coherence of the link between the most suitable habitats, the response curves and the values/categories of the corresponding environmental variables.


# SPECIFIC COMMENTS
- Please, replace "ecotour" with "ecotourism" throughout the paper.

##### ABSTRACT
- L23-24 “a presence-only (Maxent) habitat suitability model”:
I do not agree because you used one regression technique based on presence and absence or stratified pseudo-absence data (i.e., background data) (Brotons et al. 2004; Guisan et al. 2017).
- L28: “presence-only data”:
Ditto, I do not agree because you used one regression technique based on presence and absence or stratified pseudo-absence data (i.e., background data) (Brotons et al. 2004; Guisan et al. 2017).

##### INTRODUCTION
- L51: At least one reference is missing.
- L54-55: the term “especially for cryptic species” comes totally out of left field. You can rephrase this sentence by providing a little more detail and flow in the logic of your reasoning.
- L55-56: Ditto, you can rephrase this sentence by providing a little more detail and flow in the logic of your reasoning.
- L61-64: Ditto, you can rephrase this sentence by providing a little more detail and flow in the logic of your reasoning.
- L73-74: There are four references. You can delete one or two of them.
- L84-86: You can add the term “worldwide” so that it is clear that you are talking about the tropical and temperate waters of the world.
- L93: “short-term” and “long-term” instead of “short term” and “long term”, respectively.
- L95-97 “Anthropogenic influences on dolphin distribution include commercial and recreational activities, such as fishing trawlers driving bottlenose dolphin foraging to coastal waters of a small embayment (Chilvers et al. 2003).”:
What impacts does this have on dolphins? Are they negative or positive for dolphins? Be more specific.
- L97-99 “In addition, there are numerous sub-species and populations of bottlenose dolphins that display different movement patterns and home-range sizes in relation to environmental influences (Paschoalini & Santos 2020).”: this sentence comes totally out of left field. You can rephrase this sentence by using the one before and the one after in order to bring more fluidity to your reasoning, and you can insert the link in relation to the term “cryptic species”.
- L102-106: You can specify, if this is the case and I am not mistaken, that these two T. australis populations (the one in the Port Phillip Bay and the other in the Gippsland Lakes) are indeed two distinct, resident populations that do not mix (?). You can also specify if there is a spatio-temporal overlap and/or interactions of T. australis with the other two species (T. truncatus and T. aduncus) or not at all? Or is T. australis the only species of the genus Tursiops inhabiting in the Port Phillip Bay and the Gippsland Lakes?
- L107: Can you specify by whom the species was classified as ‘Critically Endangered’ and justify it with a reference?
- L112-113: Can you put a reference more recent than 2014...?
- L114-115: Ditto, can you put references more recent than 2013...?
- L115-117: It seems that you are implying the negative impact of urbanization on the presence of dolphins.
- L118-119: Can you specify what species is it? T. australis?
- L121: Ditto, can you specify what species is it? T. australis?

##### MATERIALS AND METHODS
- L147: “the spring-autumn period (September to April)”:
Since you wrote to the lines 126-129 “In southern Port Phillip Bay, there are a variety of “swim-with-dolphin” and ecotourism boat charters that have operated daily throughout the summer months for numerous decades and opportunistically recorded dolphin sightings for over a decade (Howes et al. 2012; Scarpaci et al. 2003).”, you should reverse the terms “September” and “April” on the line 147.
- L151-153 “Where multiple sightings were recorded on the same day, only those sightings with a minimum time interval of > 15 min or distance > 500 m were considered unique presence records.”:
It comes totally out of left field. Can you say why you did this and cite at least one reference? And also argue about the choice of the values of 15 minutes and 500 m?
- L163: You can delete the term “(Victorian Government)” because it has already been mentioned above (lines 149-150).
- L169 “presence-only analyses”:
I do not agree because you used one regression technique based on presence and absence or stratified pseudo-absence data (i.e., background data) (Brotons et al. 2004; Guisan et al. 2017).
- L176-178 “. Dynamic environmental variables such as sea surface temperature and sea surface chlorophyll-a are available only at low spatial resolution across Port Phillip Bay.”:
You should specify the spatial resolution of these two dynamic environmental variables and the corresponding references, please.
- L181: “are” instead of “is”.
- L187: you can delete the comma after “(TPI)”.
- L204-216: I suggest replacing and dispersing the different information from this paragraph in the previous paragraphs (L183-203) for each of the variables in order to avoid going back and forth between the different paragraphs.
You should systematically put the spatial resolution and the source/reference for each of the environmental variables used in the model: the spatial resolution for the bathymetry and the benthic habitat, and the source/reference for the slope, aspect and TPI are missing.
It would also be great if you added a one-sentence hypothesis of what you might expect before getting your results for each of the seven environmental variables used in the model in the lines 180-203 in order to ensure rigor of scientific reasoning by exposing the underlying assumptions of your scientific question.
- L220-221 “A presence-only habitat suitability model”:
I do not agree because you used one regression technique based on presence and absence or stratified pseudo-absence data (i.e., background data) (Brotons et al. 2004; Guisan et al. 2017).
- L222-224 “Prior to modelling, individual environmental raster layers were stacked into a single R object before a data frame containing extracted environmental values at presence-only data points was produced.”:
Ditto, I do not agree because you used one regression technique based on presence and absence or stratified pseudo-absence data (i.e., background data) (Brotons et al. 2004; Guisan et al. 2017). Moreover, you probably used the environmental variables also associated with the background points. Please reorganize and respect the rigor of your modelling approach.
- L231 “with benthic habitat specified as a fixed factor (categorical variable) prior to modelling.”: you can delete this since it has already been said on the lines 194-195.
- L232 “A presence-only habitat suitability model”:
I do not agree because you used one regression technique based on presence and absence or stratified pseudo-absence data (i.e., background data) (Brotons et al. 2004; Guisan et al. 2017).
- L236-237 “This approach was selected as it performs well in its predictive accuracy when true absences and search effort information are lacking (Elith et al. 2006; Phillips et al. 2004).”:
As explained and justified in the general comments, I disagree.
- L237-239 “To reduce the effect of sampling bias, background points were extracted from areas near to dolphin presences.”:
What do you mean by “areas near to dolphin presences”? Were these background points extracted only in the area corresponding to the model calibration zone (which is the zone sampled by ecotourism and other vessels of opportunity) or within the entire area covering the entire Port Phillip Bay? Indeed, to avoid any bias linked to the absence of sampling in certain areas of the Port Phillip Bay, the background points must be selected only in the areas which were sampled by ecotourism and other vessels of opportunity. Please read up on this, for example in Lasram et al. (2020), Schickele et al. (2020), Charbonnel et al. (2022), Navarro et al. (2023) and other references, to learn more about the rigor of their background (or pseudo-absence) points selection. And, please, specify the model calibration area and model projection area in the text.
- L239-240 “and 5,000 random background points were sampled.”:
-- You have n = 534 presence points. And did you use 5,000 points random background points? Charbonnel et al. (2022) explained “the much larger number of pseudo-absences compared to presences generates a very low prevalence. In order to avoid negative prevalence effects, the prevalence was then set to 0.5 to give equal weights to presence and pseudo-absence cells”. Indeed, the prevalence is equal to the proportion of presences in the dataset. Did you check the prevalence of your model? Can you precise how much it was set?
-- I suggest you also add these background points on the new future Figure 1 like Figure 1 of Navarro et al. (2023) in slightly transparent color and smaller in size compared to the presence points and behind the presence points so that these presence points are visible.
- L245-246 “and 10 Maxent models were fitted and evaluated using the random training and testing data”:
I highly recommend you read the fairly recent book "Habitat suitability and distribution models with applications in R" written by A. Guisan, W. Thuiller and N.E. Zimmermann (2017) in order to better understand how a model with your choice of partition into calibration (75%) and evaluation (25%) datasets is evaluated. Indeed, “Each single model is run on the training partition and evaluated on the test partition using the area under the ROC curve (AUC; see Part IV)” (Guisan et al. 2017), could you explain why you evaluated the model with AUC on the training (or calibration) dataset???
- L246-248 “As Maxent interprets ‘1’ as presence and ‘0’ as pseudo-absence a value of ‘1’ was assigned to presence datasets and a value of ‘0’ to background datasets.”:
I wonder if this sentence should be written since it seems logical?
- L254-257 “Diagnostic plots of commission rates and the AUC metric were used to assess model performance. An AUC score > 0.5 suggests that a model performs better than random. Model performance was determined following Hosmer at al. (1989): < 0.5 = none; 0.5-0.7 = poor; 0.7-0.8 = acceptable; 0.8-0.9 = excellent; > 0.9 = outstanding.”:
In the book "Habitat suitability and distribution models with applications in R" written by A. Guisan, W. Thuiller and N.E. Zimmermann (2017), it is mentioned that: “A curve that goes below the 1:1 line means that the model yields predictions that are worse than random, i.e. counter-predictions (similar to negative correlation coefficients), with values between 0 and 0.5. […]. Araújo et al. (2005a) proposed a refined AUC scale, with AUC > 0.90 being “excellent”; 0.80 < AUC < 0.90 being “good”; 0.70 < AUC < 0.80 “fair”; 0.60 < AUC < 0.70 “poor”; 0.50 < AUC < 0.60 “fail”, and AUC < 0.5 being “counter-predictions”.” You can use criteria from a source newer than 1989.
- L257-259 “The importance of environmental variables was determined by percentage of contribution and a jack-knife analysis of the training and testing AUC”:
-- “jackknife” instead of “jack-knife”
-- I strongly recommend researching how a jackknife test actually works for variable importance (it's not using the training and testing AUC): https://consbiol-unibern.github.io/SDMtune/articles/variable-importance.html and other references you can find, and then please correct this sentence.

##### RESULTS
- L264-268: It should be mentioned in one sentence that there is no data for the years 2018 and 2020 and explain the reasons for this.
- L268: You can add "(n=118)" after "2022".
- L275-276 “Nonetheless, the averaged Maxent model had an acceptable performance for both training (AUC = 0.8 (± 0.007)) and testing (AUC = 0.79 (± 0.01)) datasets.”:
-- I highly recommend you read the fairly recent book "Habitat suitability and distribution models with applications in R" written by A. Guisan, W. Thuiller and N.E. Zimmermann (2017) in order to better understand how a model with your choice of partition into calibration (75%) and evaluation (25%) datasets is evaluated. Indeed, “Each single model is run on the training partition and evaluated on the test partition using the area under the ROC curve (AUC; see Part IV)” (Guisan et al. 2017), could you explain why you evaluated the model with AUC on the training (or calibration) dataset???
-- Depending on the criteria you have chosen to qualify the AUC score, please use the correct term using one of those used in these criteria.
- L289-292 “Response curves present consistent negative relationships between the likelihood of dolphin occurrence, and distance to shipping channels (m) and coastline (m), with probability decreasing with increasing distance (Fig. 3 a,b). This indicates that bottlenose dolphins in Port Phillip Bay were more likely to be present in waters either closer to shipping channels or the coastline.”:
I find it difficult to follow you because you wrote “Human activities occurring in Port Phillip Bay have been suggested to influence bottlenose dolphin behavior and habitat relationships (Filby et al. 2017; Hewitt et al. 2004; Scarpaci et al. 2003). Hence, two variables reflecting the environment in relation to urbanization and vessel (commercial and recreational) activity were included as predictor variables in the model as Euclidean distance to coastline (m) and Euclidean distance to shipping channels (m), respectively.” (L196-201) and “In addition, continued coastal urban growth and industrialization in the region could have detrimental effects on large marine predators such as bottlenose dolphins (Marley et al. 2017; Zanardo et al. 2017) in Port Phillip Bay.” (L115-117), and I would have tended to think that anthropogenic activities such as urbanization and the proximity of ships would have had a negative impact on the presence of dolphins. However, here, it would seem to be the opposite since we have a negative relationship between the probability of bottlenose dolphin presence and distances to shipping channels or to coastlines (new future Figure 2). This point merits i. to have a hypothesis upstream (positive or negative impact according to the literature?) which could be added to lines 196-201 and ii. to be discussed and criticized further in discussion. In addition, you are taking the liberty of putting "consistent" on line 289 whereas you have not made an assumption about what might be expected.
- L301: you can “(Table 2)” after “dolphin presence”.
- L313-315 “The predicted areas of high suitability were in coastal waters close to Hobsons Bay (Melbourne), Sorrento and Corio Bay/Geelong.”:
You should add also “Queenscliff”.
- L315-316 “In contrast, deeper waters in the center, and shallow coastal waters in the eastern extent, of Port Phillip Bay were predicted to be of lowest habitat suitability.”:
You should also add the coast which is south of Werribee. And if it has a name, you can specify it in the text and add it to the figures (new future Figures 1 and 4).
- Table 2:
-- “Environmental variables and their contribution (mean ± SD)” instead of “Environmental variable contribution (+/- SD)”.
- Figures 1 and 2:
-- You can merge the different information from figures 1 and 2 into a single figure (new future figure 1).
-- You can add labels to the figure to make the link with the text of the article such as "Port Phillip Bay", “Bass Strait”, “Port Phillip Heads”, and add the location of shipping channels.
-- Does the pink zone “Vessel operational areas” only represent ecotourism vessel operational areas? If yes, another area should be added representing other vessels of opportunity in another color and thus modify the title of the figure.
-- I suggest you also add these background points on the new future Figure 1 like Figure 1 of Navarro et al. (2023) in slightly transparent color and smaller in size compared to the presence points and behind the presence points so that these presence points are visible.
-- You should remove “n = 534” from the figure and rephrase "n = number of sightings used in habitat suitability model" into a grammatically correct phrase in the figure title, please.
- Figure 3 (new future figure 2)
- Figure 4 (new future figure 3)
- Figure 5 (new future figure 4):
-- You can add labels to the figure to make the link with the text of the article such as "Port Phillip Bay", “Bass Strait”, “Port Phillip Heads”, and add the location of shipping channels.
-- To understand that Bass Strait is not included in the projection area, you should put this area in white to avoid any confusion given that the blue color is used for the habitat suitability index and predictions SD index.
-- You can enlarge the legends of these two indexes and add for example “0.25”, “0.50” and “0.75” in addition to the “0” and “1” for the habitat suitability index and other values for the predictions SD index.
-- Why are there very light blue holes in the same locations on both maps?

##### DISCUSSION
- L330: “presence-only habitat suitability model”:
I do not agree because you used one regression technique based on presence and absence or stratified pseudo-absence data (i.e., background data) (Brotons et al. 2004; Guisan et al. 2017).
- L344: “presence-only models”:
Ditto, I do not agree because you used one regression technique based on presence and absence or stratified pseudo-absence data (i.e., background data) (Brotons et al. 2004; Guisan et al. 2017).
- L348-349: there are too many references.
- L351: “presence-only models”:
Ditto, I do not agree because you used one regression technique based on presence and absence or stratified pseudo-absence data (i.e., background data) (Brotons et al. 2004; Guisan et al. 2017).
- L358-359: there are too many references.
- L361-362:
I do not agree because you used one regression technique based on presence and absence or stratified pseudo-absence data (i.e., background data) (Brotons et al. 2004; Guisan et al. 2017).
- L362-365: “Indeed, previous research modelling habitat suitability for Siberian Jays (Perisoreus infaustusin) in the boreal forests of Sweden found no difference in model performance predictions generated with opportunistically collected citizen science data (presence-only) and systematic surveys (presence-absence) (Bradter et al. 2021).”:
Certainly, but there are articles that proved the opposite. Please don't ignore them and discuss more on this point.
Moreover, I do not agree with you since reading the abstract of the article of Bradter et al. (2021), this is not what you mention in your text... Please pay more attention to the content of the references you mention in a research paper.
- L370-373: Could you better explain why the anthropogenic activities studied (proximity to the shipping channels and the urban areas) offer favourable foraging conditions for dolphins? We might have thought the opposite.
- L373: At least one reference is missing after “in other studies”.
- L395-396 “Interestingly, the results of the present study indicated that the deeper, central areas of Port Phillip Bay were not of high habitat suitability.”:
I suggest you rephrase this sentence since it is not possible to confirm this because ecotourism and other vessels of opportunity did not sample these areas.
- L422: Similar to what you have just written for chlorophyll-a concentration, you can add one sentence with associated reference(s) regarding what is known about the influence(s) of water temperature.

##### CONCLUSIONS
- L434: Change section title to “Conclusions”.
- You could mention also the possibility of studying the influence of other anthropogenic factors on the distribution of bottlenose dolphins, such as climate change. With associated references from studies that have proved this on dolphins in other places around the world.

##### ACKNOWLEDGMENTS
- L460-461: And the vessels of opportunity and researcher?

##### REFERENCES
- Indeed, 107 references in total is quite a lot for a rather short article, quite concise and whose problem is not very complex. You can cut the total number of references almost in half and favour references that are more recent.

##### SUPPLEMENTAL FILES
- It would be desirable to add the mapping of values/categories for each of the environmental variables used in the habitat suitability modelling across the entire study area in order to visualize the coherence of the link between the most suitable habitats, the response curves and the values/categories of the corresponding environmental variables.

---

## Round 0.2 · accepted · Accept

Dear Maddison and co-authors,

I apologize for the delay in this decision. Along with one of the reviewers, I have carefully addressed your rebuttal and revised manuscript and am please to accept it for publication in PeerJ. Congratulations and thank you for your contribution to this research field!

With warm regards,
Xavier

Reviewer 1 ·

Basic reporting

I have no further comments to the manuscript considering the changes and comments provided by the authors.

Experimental design

I have no further comments to the manuscript considering the changes and comments provided by the authors.

Validity of the findings

I have no further comments to the manuscript considering the changes and comments provided by the authors.

Additional comments

I have no further comments to the manuscript considering the changes and comments provided by the authors.